# Steroid-Induced Hyperglycemia and Its Effect on Outcomes of R-CHOP Chemotherapy for Diffuse Large B-Cell Lymphoma

Mark Kristjanson [1,2,*], Pascal Lambert [3,4], Kathleen M. Decker [3,4,5], Sonja Bruin [6], Elizabeth Tingey [7] and Pamela Skrabek [8,9]

1    Department of Family Medicine, Max Rady College of Medicine, Rady Faculty of Health Sciences, University of Manitoba, Winnipeg, MB R3E 0W2, Canada
2    Urgent Cancer Care, CancerCare Manitoba, 675 McDermot Avenue, Winnipeg, MB R3E 0V9, Canada
3    Paul Albrechtsen Research Institute CancerCare Manitoba, 675 McDermot Avenue, Winnipeg, MB R3E 0V9, Canada; plambert@cancercare.mb.ca (P.L.); kdecker@cancercare.mb.ca (K.M.D.)
4    Department of Epidemiology and Cancer Registry, CancerCare Manitoba, 675 McDermot Avenue, Winnipeg, MB R3E 0V9, Canada
5    Department of Community Health Sciences, Max Rady College of Medicine, Rady Faculty of Health Sciences, University of Manitoba, 750 Bannatyne Avenue, Winnipeg, MB R3E 0W2, Canada
6    Ongomiizwin Health Services Institute of Indigenous Health and Healing, 745 Bannatyne Avenue, Winnipeg, MB R3E 3N4, Canada; sonja.bruin@umanitoba.ca
7    Max Rady College of Medicine, Rady Faculty of Health Sciences 750 Bannatyne Avenue, Winnipeg, MB R3E 0W2, Canada; tingeye@myumanitoba.ca
8    Department of Medical Oncology & Hematology, CancerCare Manitoba, 675 McDermot Avenue, Winnipeg, MB R3E 0V9, Canada; pskrabek@cancercare.mb.ca
9    Department of Internal Medicine, Max Rady College of Medicine, Rady Faculty of Health Sciences, University of Manitoba, Winnipeg, MB R3E 0W2, Canada
*    Correspondence: mkristjanson@cancercare.mb.ca

**Abstract:** Large doses of steroids are integral to R-CHOP, a first-line systemic therapy for diffuse large B-cell lymphoma (DLBCL), an aggressive form of non-Hodgkin Lymphoma (NHL). Patients on R-CHOP often develop clinically significant hyperglycemia from steroids. There is evidence of harms from steroid-induced hyperglycemia in the context of chemotherapy which are associated with a reduction in overall survival. The objective of our study was to characterize the effect of steroid-induced hyperglycemia on the outcomes of R-CHOP chemotherapy for DLBCL. Methods: We performed a retrospective chart review of 188 patients with DLBCL treated with R-CHOP through CancerCare Manitoba (CCMB) from 1 January 2010 to 31 December 2014. Patients diagnosed with DLBCL were identified using the Manitoba Cancer Registry. The CCMB electronic medical record was reviewed to examine the association between steroid-induced hyperglycemia and subsequent infection, including febrile neutropenic events and overall survival (OS). Results: Patients who developed hyperglycemia with steroid exposure became hyperglycemic during their first R-CHOP cycle. No significant differences in OS or rates of infection were found between euglycemic and hyperglycemic subjects. Conclusions: Patients destined to develop steroid-induced hyperglycemia declare themselves early in the course of steroid exposure. No statistically significant reduction in overall survival attributable to steroid-induced hyperglycemia was found.

**Keywords:** lymphoma; steroid; hyperglycemia; toxicities

## 1. Introduction

Endogenous glucocorticoids such as cortisol, and synthetic steroids such as prednisone, promote increased glucose production (via the stimulation of proteolysis and increased hepatic gluconeogenesis) and decreased tissue uptake of glucose via the inhibition of peripheral glucose transport and utilization [1]. An association between the administration of exogenous glucocorticoids and the development of hyperglycemia is well documented [2–4].

The high doses of steroids used in chemotherapy regimens for the treatment of leukemias, lymphomas, and multiple myeloma can prove especially problematic in this regard. Steroid-induced hyperglycemia has been associated with an increase in treatment-related toxicities including neutropenia, infection/sepsis [2,4,5], and chemotherapy-related neuropathies [6]. Associations have also been documented between steroid-induced hyperglycemia and a greater risk of failing to complete the intended course of chemotherapy, early disease relapse, and decreased overall survival [2–4].

R-CHOP is a first-line chemotherapy regimen used in the curative-intent treatment of diffuse large B-cell lymphoma (DLBCL). The R-CHOP regimen (rituximab, cyclophosphamide, doxorubicin, vincristine, prednisone) used in Manitoba includes prednisone 100 mg po daily for days 1–5 in a 21-day cycle. Steroids are part of lymphoma treatment regimens as they stimulate lymphocytes to undergo apoptosis. Exposure to high doses of steroids can cause or exacerbate hyperglycemia; this may be to a degree that negatively impacts patients' health. The hyperglycemia associated with the high doses of steroids used in some chemotherapy regimens has been shown to negatively impact treatment outcomes [2–4]. In hematological malignancy treated with high-dose steroids as part of Hyper-CVAD, which uses much larger steroid doses than those used in R-CHOP, steroid-related hyperglycemia has been associated with lower response rates and inferior overall survival [2]. On the basis of a small number of retrospective studies, the incidence of steroid-induced hyperglycemia in patients on R-CHOP has been estimated to be nearly 30% [3,4,6]. In a study by Lamar et al., steroid-induced hyperglycemia during R-CHOP or DA-EPOCH-R chemotherapy for non-Hodgkin lymphoma was associated in univariate analysis with reduced overall survival. Multivariate analysis did not show a significant association between hyperglycemia per se and reduced survival, but it did reveal a significant association of hyperglycemia to chemotherapy dose alteration, which in turn, was significantly associated with reduced OS [3]. Another study by Zhou et al., likewise shows a reduction in PFS and OS in subjects who develop steroid-induced hyperglycemia in the context of therapy with R-CHOP for DLBCL. Of note, the study by Zhou et al., also suggests that the adverse effects of steroid-induced hyperglycemia on PFS and OS can be mitigated with the use of metformin [5]. The objective of our study was to characterize the effect of steroid-induced hyperglycemia on outcomes of R-CHOP chemotherapy for individuals diagnosed with diffuse large B-cell lymphoma (DLBCL) in Manitoba, Canada.

## 2. Materials and Methods

### 2.1. Cohort

This study used a retrospective cohort design. Individuals age 18 and older who were diagnosed with non-Hodgkin lymphoma in Manitoba from 1 January 2010 to 31 December 2014 were identified in the Manitoba Cancer Registry (MCR) using (ICD-0-3) code 9591/3. The MCR was used to determine stage at diagnosis, age at diagnosis, and date of death. We reviewed 188 charts out of a total of 386 available charts from this cohort. The CCMB electronic medical record (ARIA) was used to identify from this cohort of NHL patients those individuals diagnosed with DLBCL per se who received R-CHOP; a chart review was undertaken to identify the details of their receipt of R-CHOP including the dates of each chemotherapy and medication doses for each cycle, all blood glucose measurements from the start of treatment to final treatment, any additional steroid use and dose, history of diabetes, other co-morbidities, and complications of treatment, both hematologic (e.g., febrile neutropenia) and non-hematologic (e.g., peripheral neuropathy). Hospital abstracts data were used to identify any in-patient hospitalizations during the treatment period.

### 2.2. Analysis

The study cohort was described using frequencies, percentages, means, and standard deviations. Latent class mixed models and joint latent class mixed models were used to create trajectory groups of blood glucose levels during chemotherapy, which represent clusters of individuals based on their glucose values during follow-up. Note that one

advantage to this approach is that the use of mixed models allows the inclusion of unequal interval times. Due to the study cohort size, the analyses included a maximum of two trajectory groups. Latent models used an automatic grid search function (50 sets of initial values and 10 iterations) to reduce the odds of a convergence towards a local maximum. Model fits were evaluated with relative entropy and odds of correct classification. Joint models were used to associate the time-varying blood glucose variable to time-to-event outcomes (i.e., infections). For all mixed models, splines were used to account for non-linear time effects for both fixed and random effects. Kaplan–Meier curves with a one-year landmark were used to compare survival between blood glucose trajectory groups during the first year period post chemotherapy initiation. Cox regression models predicting infection and response to chemotherapy with a one-month landmark were used to compare the 1-month blood glucose average during the remaining 11-month period of the first year of follow-up since chemotherapy initiation. The potential non-linear relationship was accounted for with splines. Predicted probabilities during follow-up were plotted for glucose values of 5, 7.5, and 10 mmol/L. Analyses were run using R version 4.1.3 [7] with the following packages: ggplot2 [8] lcmm [9,10], JM [11], survival [12,13] lmtest [14], and sandwich [15]. Note that in our study, we used the trajectory groups to delineate a "hyperglycemic" group from a "euglycemic" group, rather than specifying in advance a fixed blood glucose threshold above which a patient would be deemed to have steroid-induced hyperglycemia. Please see in Figure 1 the mean blood glucose levels over time for each of the two trajectory groups.

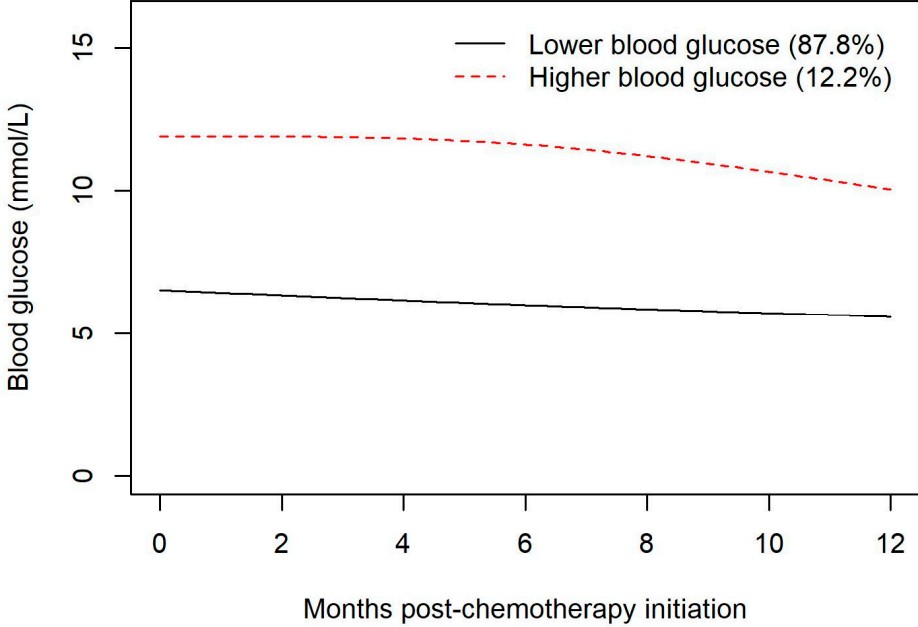

**Figure 1.** Mean blood glucose levels (mmol/L) for two distinct clusters (a euglycemic group and a hyperglycemic group) by month during the first year of follow-up post chemotherapy initiation.

## 3. Results

During the first year of follow-up, individuals diagnosed with DLBCL (*n* = 188) had a mean age of 62 years, and 85% were intended to receive six cycles of R-CHOP. Patients had an average of nine blood glucose measurements during therapy. Eleven percent of the cohort completed less than the intended number of R-CHOP cycles primarily due to toxicities.

### 3.1. Blood Glucose and Survival

A latent class mixed model was used to create trajectory groups of blood glucose levels during the first year of follow-up post chemotherapy initiation. Those trajectory groups

were included in a Kaplan–Meier to compare survival during the follow-up starting one year post chemotherapy initiation. The latent class model created two trajectory groups and model fits indicated a relative entropy of 0.97 and odds of correct classification of 235 and 33, indicating a high expectation of accuracy. The two trajectory groups created were higher (hyperglycemic) and lower (euglycemic) glucose groups. The majority of patients remained euglycemic (87.8%) despite the receipt of high doses of steroids used in R-CHOP; 12.2% of the group had early and sustained hyperglycemia (see Figure 1). Diabetes was included as a class membership variable, and indicated that those with diabetes had a significantly higher risk of belonging to the hyperglycemia group (OR = 30.71; 95% CI: 9.96–94.68; $p < 0.001$). The two groups demonstrated a non-significant difference ($p = 0.589$) in OS in the years following the first year post chemotherapy initiation (see Figure 2).

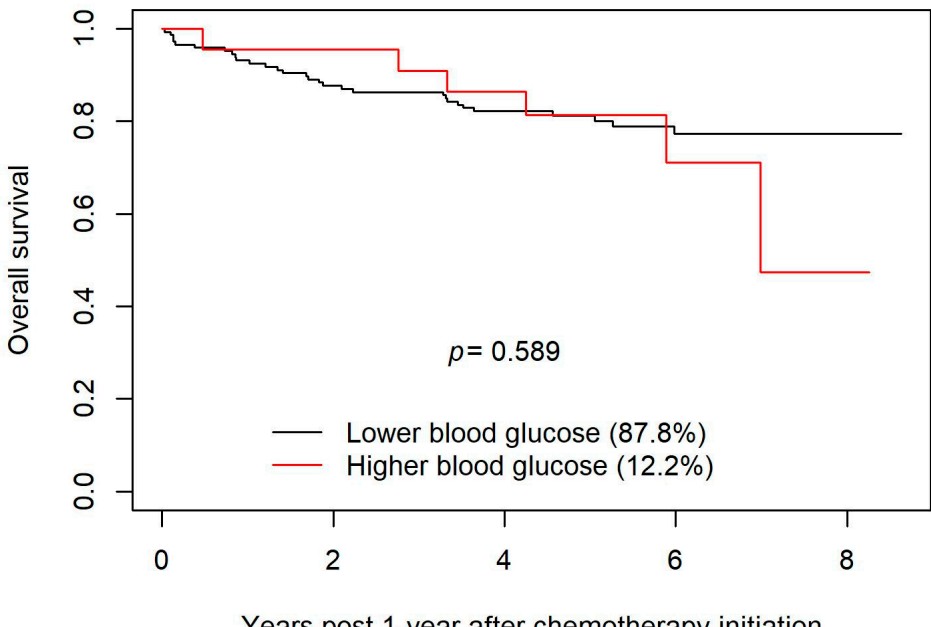

**Figure 2.** Overall survival between the euglycemic group and the hyperglycemic group by year 1 year post chemotherapy initiation.

### 3.2. Blood Glucose and Infection

Of the cohort of 188 individuals, 164 individuals had blood glucose levels measured prior to infection (if they had an infection complicating their chemotherapy) or within one year of follow-up (if they had no infection). Data on this group of individuals were included in a joint latent class mixed model to create trajectory groups of blood glucose and associate the groups to the time-to-event outcome of chemotherapy-associated infection. The joint latent class model created two trajectory groups. Model fits indicated a relative entropy of 0.98 and an odds of correct classification of 525 and 84, indicating a high expectation of accuracy. The two trajectory groups created were higher (hyperglycemic) and lower (euglycemic) glucose groups, representing 11% and 89% of the cohort (see Figure 3). Again, a majority of the patients maintained euglycemia. A smaller group was characterized by hyperglycemia early in the follow-up period; those who were hyperglycemic early in the follow-up period tended to remain so over the first year. Seven percent of the total had a random glucose >11 mmol/L prior to their first cycle of R-CHOP. Throughout the 12 months following the initiation of R-CHOP, patients who tended to hyperglycemia were at greater risk of infection, although this was not significant (HR 1.79, 95% CI = 0.76–4.17) (see Figure 4).

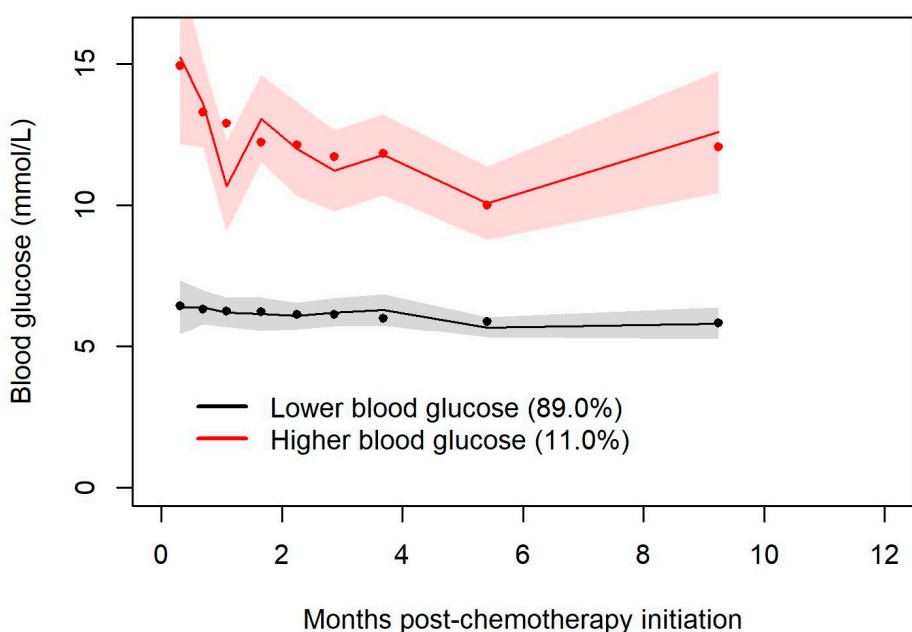

**Figure 3.** Blood glucose levels (mmol/L) for two distinct clusters (a euglycemic group and a hyperglycemic group) by month during the first year of follow-up post chemotherapy initiation.

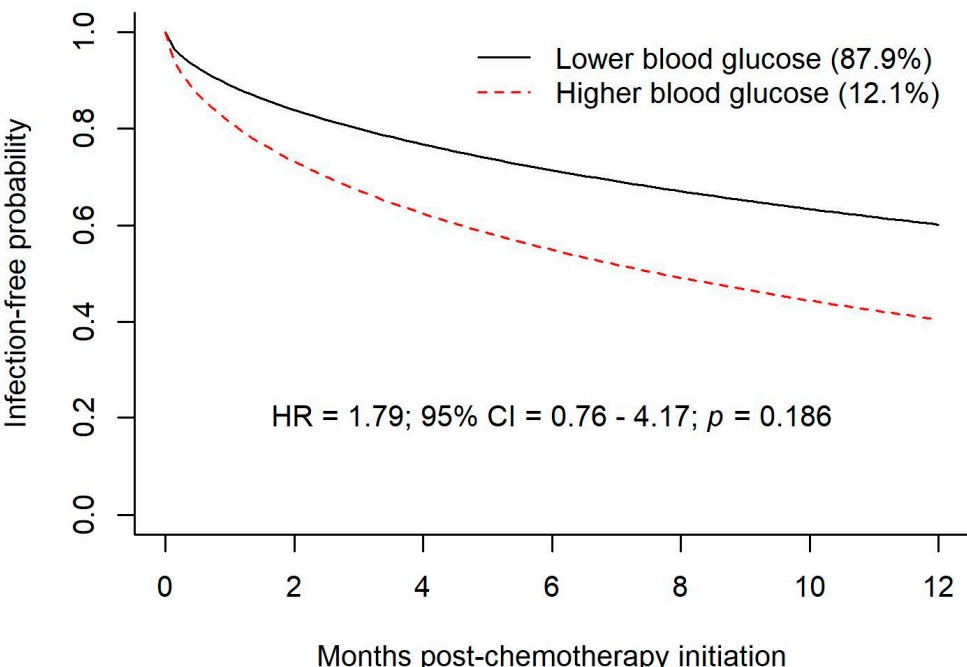

**Figure 4.** Predicted infection—free probabilities by trajectory groups (association parameter from joint latent class mixed model.

One-hundred and forty-three individuals had a blood glucose measurement that was obtained a) during their first month of follow-up and b) prior to having an infection (for those individuals who did in fact have an infection at a subsequent time). These subjects were included in an analysis where the average of the blood glucose values during the first month of chemotherapy was included as a predictor in a Cox regression model predicting infection. A non-significant increased risk of infection as blood glucose increased was found ($p = 0.259$). Predicted infection-free probabilities for blood glucose values of 5, 7.5, and 10 mmol/L are included in Figure 5. Although the differences were not statistically

significant, data trends suggested that the risk of subsequent infection associated with early hyperglycemia increased as the average blood glucose level increased.

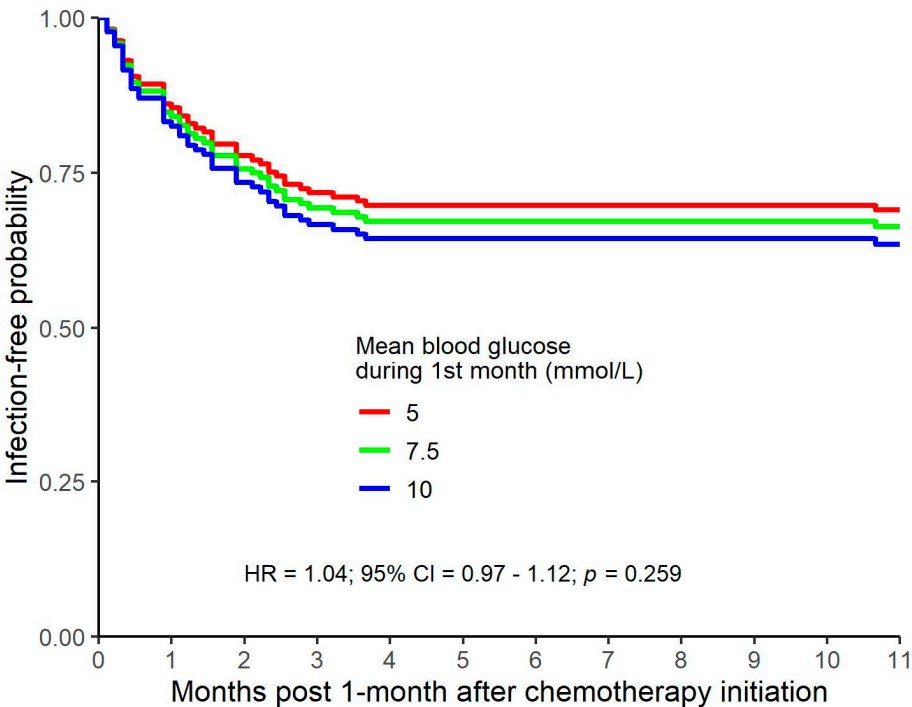

**Figure 5.** Predicted infection-free probabilities after mean blood glucose during first month of follow-up.

Using 164 individuals that had blood glucose levels measured prior to infection (if they had an infection) or within one year of follow-up (if they had no infection), joint models were used to associate the time-varying blood glucose values with the risk of infection. When associating the blood glucose value at the time of event, a 16% increased risk of infection was found with higher blood glucose values (HR 1.16, 95% CI 1.03–1.32, *p* = 0.017; see Table 1). This association attenuated when values earlier than that at the time of event were associated with the outcome (1 month prior: HR 1.15, 95% CI 1.01–1.30, *p* = 0.034; 2 months prior: HR 1.13, CI 0.99–1.28 *p* = 0.064; see Table 1).

**Table 1.** Joint model with time-varying predictor of blood glucose predicting infection.

| Temporal Relation of Blood Glucose to Time of Infection | HR | 95% CI | *p* |
|---|---|---|---|
| Blood glucose at time of infection | 1.16 | 1.03–1.32 | 0.017 |
| Blood glucose 1 month prior to infection | 1.15 | 1.01–1.30 | 0.034 |
| Blood glucose 2 months prior to infection | 1.13 | 0.99–1.28 | 0.064 |

## 4. Discussion

In our study, blood glucose levels and the risk of infection were investigated with multiple approaches.

- Trajectory groups of blood glucose values demonstrated a non-significantly increased risk of infection during chemotherapy for hyperglycemic patients.
- Higher mean blood glucose values during the first month of follow-up were non-significantly related to a higher risk of infection.
- A joint model with a time-varying predictor of blood glucose values indicated that higher blood glucose values were significantly related to a higher risk of infection when closer to time of infection.

In our study, the increased risk of infection associated with hyperglycemia at the time of infection attained statistical significance when assessed with a joint model with a time-varying predictor of blood glucose values. One would expect a temporal and clinically significant association between hyperglycemia and infection for at least two reasons. Firstly, a causal relationship would be expected because hyperglycemia has an adverse effect on neutrophil function concurrent with hyperglycemia. A second causal association, with the direction of causality reversed, would be anticipated because an active infection and the stress response associated with infection tends to exacerbate hyperglycemia in individuals who already have some degree of glucose intolerance. The association between an instance of hyperglycemia and the observed risk of subsequent infection was still significant for blood glucose readings obtained one month prior to the infection but lost statistical significance with greater temporal separation. Although one method of analysis of our data does suggest a higher risk of infection around the time of hyperglycemia, this association could be random, given that other methods of analysis do not show a significant association between glucose levels and the risk of infection. A plausible alternative explanation is that analysis of data from a larger study cohort would demonstrate an increase in the risk of infection in patients who are hyperglycemic.

One strength of our study arises from the use of trajectory groups to characterize the temporal association between steroid exposure and the development of hyperglycemia. In our study, the creation of trajectory groups demonstrates that patients who are destined to develop steroid-related hyperglycemia do so very early in the course of their exposure to steroids and tend to experience sustained hyperglycemia throughout the period of steroid exposure. This observation highlights the need for clinicians to take note of hyperglycemia that emerges early in the course of systemic therapy and to ensure ongoing monitoring of blood glucose levels for such individuals. It suggests that clinicians might beneficially incorporate into the routine monitoring of therapy with R-CHOP a non-fasting blood glucose determination during the first cycle of R-CHOP at some point during cycle days 1–5.

Our study looked retrospectively at patients who were on R-CHOP, which entails significant exposure to steroids in the form of dexamethasone and large doses of prednisone. This study adds to the small but growing body of literature that seeks to clarify whether patients who develop steroid-related hyperglycemia on R-CHOP suffer greater toxicities and/or have shorter overall survival. The findings of our study are reassuring insofar as the absence of a statistically significant reduction in OS suggests that any adverse impact of steroid-induced hyperglycemia on R-CHOP outcomes is small. Likewise, we sought evidence that hyperglycemia confers an increased risk of infection; two of the three methods of statistical analysis employed to this end did not find a significant increase in the risk of infection in those patients who developed hyperglycemia. Approximately 120 new cases of DLBCL are diagnosed in Manitoba each year. An audit of the charts of all the individuals diagnosed with DLBCL during the period examined in our study would have entailed the audit of approximately 600 charts. Although the cohort examined in our study is large enough to have detected any large differences in the outcomes of R-CHOP, examination of a much larger cohort might well have detected small but statistically significant differences in complication rates or OS. Thus, the main weakness of this study derives from the limitation of our audit to 188 charts. This was an observational study and therefore may be prone to bias from unrecognized or unmeasured factors. In addition, it is possible that some analyses may have been underpowered to detect a significant difference between the groups and because the study took place in Manitoba, the results might not be generalizable to other populations. To this point, it may be relevant that approximately 12% of our study cohort fell into the "hyperglycemic" group upon latent group analysis. By contrast, of the cohort examined in the study by Lamar et. al., 27% had a history of DM prior to the initiation of chemotherapy, and 47% of the study subjects had at least one blood glucose in the hyperglycemic range [3]. Although the methodologic differences preclude a head-to-head comparison between the two populations in this regard, it might be the case that the cohort examined in our study had fewer of those toxicities towards which

hyperglycemia predisposes an individual on account of generally better glycemic control in our group. The failure in our study to demonstrate a statistically significant difference in OS survival between the hyperglycemic and euglycemic subjects also limited the ability to conduct meaningfully the additional analyses that could have been performed to shed light on the reason for the difference in OS (found in the study by Lamar et. al., but not confirmed in ours) between euglycemic and hyperglycemic subjects. For example, one of the many strengths of the study by Lamar et. al. was the elucidation using multivariate regression analysis of the distinction of an apparent association between univariate analysis of hyperglycemia per se and a reduction in OS from the more substantive and probably causal association between hyperglycemia and an increased risk of failure to complete the intended course of chemotherapy, which in turn was associated with reduced OS. Further study with a larger cohort would be required to clarify this.

A small number of studies in the literature suggest harms for those patients who develop hyperglycemia on R-CHOP, including a greater risk of hematologic toxicities such as neutropenia and febrile neutropenic events, and non-hematologic toxicities such as peripheral neuropathies [3,6]. Notwithstanding the (limited) data showing such harms related to steroid-induced hyperglycemia, there is no widely accepted glycemic goal in the current standard of care for patients who develop steroid-induced hyperglycemia on R-CHOP.

A 2004 study by Weiser, Cabanillas, Konopleva et.al. examined the incidence of hyperglycemia during induction chemotherapy for ALL using Hyper-CVAD (hyperfractionated cyclophosphamide, vincristine, doxorubicin, and dexamethasone) alternating with MTX/Ara-C (methotrexate-cytarabine) and the relationship between the duration of remission and hyperglycemia. Hyper-CVAD includes high dose dexamethasone–viz., 40 mg daily on days 1–4 and 11–14. MTX/Ara-C includes intravenous methylprednisolone 50 mg twice daily on days 1–3. Their study found that 37% of patients had two or more blood glucose readings $\geq$ 200 mg/dL (11.7 mmol/L) and that those patients had a shorter median survival than did euglycemic patients (29 months vs. 88 months; $p < 0.001$). The hyperglycemic cohort was more likely to develop sepsis and other complicated infections compared to patients without hyperglycemia [2]. A meta-analysis by Alenzi and Kelley in 2017 concluded that the odds of having chemotherapy-induced neutropenia were 32% higher among those patients who either had a pre-existing diagnosis of diabetes or who developed steroid-induced hyperglycemia during chemotherapy, compared with euglycemic patients [4]

In a study published in 2018, Zanetta Lamar et al. found that steroid-associated hyperglycemia during chemotherapy for NHL was associated significantly ($p < 0.001$) with chemotherapy dose delays and reductions, which was associated with a decrease in overall survival. However, multivariate analysis controlling for the influence of chemotherapy alteration on survival found that hyperglycemia per se was not associated significantly with reduced survival. Lamar's group examined NHL treated either with R-CHOP or DA-EPOCH-R. Of note, patients treated with DA-EPOCH-R, a regimen with higher steroid doses than those used in R-CHOP, had a higher incidence of both hyperglycemia and associated chemotherapy dose delays and/or reductions than did patients in the R-CHOP group.

Our study included patients who had received R-CHOP for DLBCL. The discrepancy between the findings of the study conducted by Lamar et al. and our study might be attributable in part to the inclusion in the study by Lamar et al. of patients treated with DA-EPOCH-R, a regimen that entails higher steroid doses than those used in R-CHOP, as well as the putatively lower incidence of hyperglycemia in the cohort examined in our study (vide supra). The noted discrepancy might also reflect the fact that both studies involved relatively small numbers of patients (~200 in each), and the possibility that stochastic effects produced an apparent difference that would be erased if data from larger cohorts were analyzed. However, a more probable explanation is that the signal suggesting in our study a possible decrease in survival with higher blood glucose values (vide supra the Kaplan–Meier curves in Figure 2) reflects a real effect that would attain statistical significance if our

study cohort had been larger. (The study by Lamar et al. suggests that if real, such an effect is likely mediated by an association between hyperglycemia and failure to complete the intended course of chemotherapy.)

## 5. Conclusions

Research conducted by other groups shows an adverse impact of hyperglycemia on chemotherapy outcomes, and there is reason on theoretical grounds to suspect that there might be such a relationship. Furthermore, if there is indeed a deleterious effect of hyperglycemia on the rates of complications of R-CHOP or on OS, it would be important to ascertain whether there is a glycemic threshold (e.g., 15 mmol/L) above which patients would experience much more clinically and statistically significant harms, and below which blood sugars should be maintained during chemotherapy. In this study, most patients were euglycemic; however, 12.2% of patients had early and sustained hyperglycemia, showing that this is not uncommon. This study does not confirm an adverse effect of hyperglycemia on the risk of infection or on overall survival. The results of our study should be interpreted in the context of its strengths and limitations. Additional research on this topic with larger study cohorts and greater statistical power is needed to clarify the nature and degree of harms associated with steroid-induced hyperglycemia in patients on R-CHOP chemotherapy for DLBCL, and to define, if possible, a glycemic threshold above which harms are especially likely, and below which blood glucose levels should be maintained during treatment with R-CHOP.

**Author Contributions:** The following individuals contributed to the various facets of authorial creative work as follows: conceptualization—S.B., K.M.D., P.S., M.K. and P.L.; methodology—P.L.; formal analysis—P.L. and K.M.D.; investigation (data collection/chart review)—E.T.; writing—original draft preparation, M.K.; writing—review and editing, K.M.D., P.L. and P.S. All authors have read and agreed to the published version of the manuscript.

**Funding:** This research received no external funding.

**Institutional Review Board Statement:** This study was conducted in accordance with the Declaration of Helsinki, and approved by the Health Research Ethics Board of the University of Manitoba (HS19575, H2016:117 approval renewed 24 February 2023).

**Informed Consent Statement:** Insofar as this study consisted only of a retrospective chart review and did not involve directly any human subjects, informed consent was not sought for this study.

**Data Availability Statement:** The data used in this analysis are owned by the government of Manitoba. We were given permission to use the data to conduct the analysis. However, we do not have permission to share the data. Researchers interested in replicating our results can apply for data access to the Provincial Health Research Privacy Committee (PHRPC), and CancerCare Manitoba. Instructions can be found at: https://www.rithim.ca/phrpc-overview (accessed on 8 June 2016) and https://www.cancercare.mb.ca/Research/research-office (accessed on 17 February 2017). The authors did not have special access privileges and interested researchers would be able to access the data in the same manner as the authors.

**Acknowledgments:** The authors acknowledge with gratitude the contributions of Jasmine Tingey to the process of chart review and data collections, and of Lin Xue in facilitating access to data. Mark Kristjanson is particularly grateful to Kathleen Decker, Pascal Lambert, and Pamela Skrabek for their invaluable mentorship and boundless patience, and for the generous contribution of their time and expertise to this project. Mark Kristjanson wishes to acknowledge with gratitude the encouragement and wise guidance of our study coordinator, Ili Slobodian, and of Ruth Loewen, the Director of CancerCare Manitoba's Community Oncology Program.

**Conflicts of Interest:** The authors declare no conflict of interest.

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
