# Peer review of "Steroid-Induced Hyperglycemia and Its Effect on Outcomes of R-CHOP Chemotherapy for Diffuse Large B-Cell Lymphoma"

_curroncol, doi:10.3390/curroncol30120738_

Round 1

Reviewer 1 Report

Comments and Suggestions for Authors

In this manuscript, the authors conducted an in-depth investigation into the relationship between steroid-induced hyperglycemia and treatment outcomes in patients diagnosed with diffuse large B-cell lymphoma (DLBCL) who were administered R-CHOP chemotherapy. Notably, the study's findings revealed no discernible disparities in overall survival or infection rates between patients who developed hyperglycemia and those who did not.

While the study is well-structured and employs suitable methodologies to address the research question, several points and recommendations for enhancement should be considered:

1.       In the abstract, it would be beneficial for the authors to provide a clear definition of 'the two groups.' This clarification will offer readers a more precise understanding of the research's focus from the outset.

2.       To enhance clarity and transparency, the authors should offer additional details about how they defined "steroid-induced hyperglycemia." Specifically, they should specify whether a specific blood glucose cutoff value or a defined increase in blood glucose levels from baseline was used as the criterion.

3.       In the section describing the statistical models used, the authors should delve into more comprehensive explanations. For instance, they should elaborate on the specific type of latent class mixed models employed and elucidate the methodological rationale behind selecting the number of latent classes.

4.       The paper suggests that larger study cohorts are necessary to detect smaller effects. This essential point should be emphasized further to underscore the importance of expanding the study population. Moreover, the manuscript should discuss the implications of the study's current limitations more explicitly, offering insights into potential biases or uncertainties that readers should consider when interpreting the results.

By addressing these recommendations, the manuscript will become more precise, transparent, and impactful, ensuring that the research findings are effectively communicated to a broader audience.

Comments on the Quality of English Language

The manuscript is generally well-written, with clear and concise language. The structure follows a logical flow, leading the reader from the introduction to the conclusion.

Author Response

  1. In the abstract, it would be beneficial for the authors to provide a clear definition of 'the two groups.' This clarification will offer readers a more precise understanding of the research's focus from the outset.

I have replaced "No significant differences in OS or rates of infection were found between the two groups” with “No significant differences in OS or rates of infection were found between euglycemic and hyperglycemic subjects”. 

  1. To enhance clarity and transparency, the authors should offer additional details about how they defined "steroid-induced hyperglycemia." Specifically, they should specify whether a specific blood glucose cut-off value or a defined increase in blood glucose levels from baseline was used as the criterion.

I have noted in the Introduction that “On the basis of a small number of retrospective studies, the incidence of steroid-induced hyperglycemia in patients on R-CHOP has been estimated to be nearly 30% [2, 5].”  I have further added, at the end of the Analysis section, the following statement: “Note that in our study, we used the trajectory groups to delineate a “hyperglycemic” group from a “euglycemic” group, rather than specifying in advance a fixed blood glucose threshold above which a patient would be deemed to have steroid-induced hyperglycemia. Please see in figure 1 the mean blood glucose levels over time for each of the two trajectory groups.”

  1. In the section describing the statistical models used, the authors should delve into more comprehensive explanations. For instance, they should elaborate on the specific type of latent class mixed models employed and elucidate the methodological rationale behind selecting the number of latent classes.

Our statistician, Pascal Lambert, notes in response to point #3 the following:

The Methods section identifies the use of latent class mixed models, and specifically identifies the lcmm R package. Identifying it as a latent class mixed model distinguishes it from other latent class analyses, such as growth mixture models in MPlus, group-based trajectory modeling from Nagin, and k-means clustering, because none of them are using mixed model approaches. The use of mixed models allows the inclusion of unequal interval times, which is something that the other approaches don't allow. Also, the sample size limited the number of groups recommended to be detected (500 or more individuals are typically required for this analysis), which is why no more than 2 groups were considered. Pascal made reference to model fit diagnostics to indicate whether there was good fit with 2 groups (e.g., relative entropy, odds of correct classification)

Please let me know if Pascal’s response addresses in satisfactory fashion the concern raised regarding latent class mixed models. I have added into the text in the Analysis section the following statement:  "Note that one advantage to this approach is that the use of mixed models allows the inclusion of unequal interval times."

  1. The paper suggests that larger study cohorts are necessary to detect smaller effects. This essential point should be emphasized further to underscore the importance of expanding the study population. Moreover, the manuscript should discuss the implications of the study's current limitations more explicitly, offering insights into potential biases or uncertainties that readers should consider when interpreting the results.

In this respect I have added to the Discussion the following comment: To this point, it may be relevant that approximately 12% of our study cohort fell into the “hyperglycemic” group on latent group analysis.  By contrast, of the cohort examined in the study by Lamar, et. al., (27%) had a history of DM prior to initiation of chemotherapy and 47% of the study subjects had at least one blood glucose in the hyperglycemic range [2].  Although the methodologic differences between our study and that of Lamar et. al. preclude a head-to-head comparison between the two populations in this regard, it might be the case that the cohort examined in our study had fewer of those toxicities towards which hyperglycemia predisposes an individual on account of generally better glycemic control in our cohort.  The failure in our study to demonstrate a statistically significant difference in OS survival between the hyperglycemic and euglycemic subjects also limited the ability to conduct meaningfully the additional analyses that could have been performed to shed light on the reason for the difference in OS (found in the study by Lamar et. al., but not confirmed in ours) between euglycemic and hyperglycemic subjects. For example, one of the many strengths of the study by Lamar, et. al., was the elucidation by multivariate regression analysis of the distinction between an apparent association on univariate analysis of hyperglycemia per se with a reduction in OS  from the more substantive and probably causal association between hyperglycemia and an increased risk of failure to complete the intended course of chemotherapy, which in turn was associated with reduced OS.  Further study with a larger cohort would be required to clarify this.

Reviewer 2 Report

Comments and Suggestions for Authors

The authors present a retrospective study analyzing the effect of steroid-induced hyperglycemia on overall survival in patients with diffuse large B-cell lymphoma treated with standard chemotherapy regimen (R-CHOP). Their results showed no significant differences in overall survival or infection rates that could be attributable to steroid-induced hyperglycemia. In conclusion, the authors mentioned the main limitation of the study- a relatively small number of patients (188 patients) and the possibility that the results might be different in a larger cohort. They also point out the importance of ascertaining a glycemic threshold above which  patients receiving chemotherapy might experience significant harm.

The topic is clinically relevant and addresses a relatively controversial problem with discordant results in the studies published so far. The investigation has been appropriately structured and conducted, with the support of statistical analysis. Figures appropriately illustrate the data.

The only suggestion concerns another relevant study conducted on a large cohort and published recently that might deserve to be mentioned: Zhou W et al, Diabetes Metab Syndr Obes. 2022; 15:2039-2049. Influence of hyperglycemia on the Prognosis of Patients with Diffuse Large B Cell Lymphoma.

Also, in some of the references listed the year of the publication is missing (refs 2, 4)

Author Response

  1. The only suggestion concerns another relevant study conducted on a large cohort and published recently that might deserve to be mentioned: Zhou W et al, Diabetes Metab Syndr Obes. 2022; 15:2039-2049. Influence of hyperglycemia on the Prognosis of Patients with Diffuse Large B Cell Lymphoma.

Thank you for this helpful recommendation.  I have added to the citations regarding the literature showing adverse impacts of hyperglycemia on therapy for lymphoma the study by Zhou, et al, and have added to the Introduction the following comment: "Another study by Zhou, et al likewise shows a reduction in PFS and OS in subjects who develop steroid- induced hyperglycemia in the context of therapy with R-CHOP for DLBCL. Of note, the study by Zhou, et al also suggests that the adverse effects of steroid-induced hyperglycemia on PFS and OS can be mitigated by the use of metformin [3]."

Thank you also for pointing out my omission of the year of publication of references 2 & 4 (now 2 & 5).  Those years of publication have now been added (2018 and 2017 respectively).

Reviewer 3 Report

Comments and Suggestions for Authors

Thank you for the authors about this paper. 

However, some considerations remain to be clarified:

-As the paper includes only DLBCL, I think that the title should show also this. Thus change from Non-Hodgkin Lymphoma to diffuse large B-cell lymphoma is required.

-The amoung of predniso(lo)ne in R-CHOP should be written, not all readers are lymphoma oncologists.

-In the Cohort paragraph should be clarified is the population NHL or DLBCL. 

-Does your hospital measure glucose levels from every patient? Thus from how many patients do you have this value? Are also the values taken bed-side included in the analysis (these are most commonly written somewhere else in the medical records)?

-Do you have information about the C-GSF use and the age range of the patients ? These could be explaining factors as there were no differences in infection rate. According to the guidelines age over 65 or diabetes could be reasons for the primary use of prophylaxis.

-The amount of diabetic patients should be clarified in the whole text. What were the percentages of these patients in the study groups?

Author Response

-As the paper includes only DLBCL, I think that the title should show also this. Thus change from Non-Hodgkin Lymphoma to diffuse large B-cell lymphoma is required.

Agreed – please see the amended title of the paper (and please let me know if the manuscript fails to make its way to you - I can't tell if the Word document has in fact attached to this Reply).

-The amoung of predniso(lo)ne in R-CHOP should be written, not all readers are lymphoma oncologists.

The following sentence has been added to the Introduction:

"The R-CHOP regimen (rituximab, cyclophosphamide, doxorubicin, vincristine, prednisone) used in Manitoba includes prednisone 100 mg po daily for days 1-5 in a 21-day cycle."

-In the Cohort paragraph should be clarified is the population NHL or DLBCL. 

I understand the nature of your concern.  In the Cohort paragraph, we describe how we “pared down” the charts selected on the basis of a diagnosis of NHL to include only those patients who have DLBCL per se.  The statement currently reads:

“Individuals age 18 and older who were diagnosed with Non-Hodgkin lymphoma in Manitoba from January 1, 2010 to December 31, 2014 were identified in the Manitoba Cancer Registry (MCR) using (ICD-0-3) code 9591/3. The MCR was used to determine stage at diagnosis, age at diagnosis, and date of death. We reviewed 188 charts out of a total of 386 available charts from this cohort. The CCMB electronic medical record (ARIA) was used to identify from this cohort individuals with DLBCL who received R-CHOP;”

I can change the last sentence in that statement to read:

The CCMB electronic medical record (ARIA) was used to identify from this cohort of NHL patients those individuals diagnosed with DLBCL per se who received R-CHOP.

Please let me know if that appropriately addresses your concern.

-Does your hospital measure glucose levels from every patient? Thus from how many patients do you have this value? Are also the values taken bed-side included in the analysis (these are most commonly written somewhere else in the medical records)?

The vast majority of our DLBCL patients receive their first and all subsequent cycles of R-CHOP on an out-patient basis. Serum chemistry is drawn at baseline and prior to each cycle; that bloodwork typically includes a random blood glucose level, but there is no explicitly articulated requirement of the haematologist to do so.  Those lab values are all stored in the electronic medical record used at CCMB, namely ARIA.  The outpatient visits conducted throughout treatment with R-CHOP do not include point-of-care blood glucose determinations; rather, if the haematologist wants to record the patient’s blood glucose, the patient is furnished with a requisition (for serum chemistry, including a glucose level) and they are sent to the lab.

I have not at this point modified the text in the article regarding this point; please let me know if you think I should add explicit comment on this matter.

-Do you have information about the C-GSF use and the age range of the patients? These could be explaining factors as there were no differences in infection rate. According to the guidelines age over 65 or diabetes could be reasons for the primary use of prophylaxis.

This is an excellent point - that could provide additional insight into why infection rates were not statistically significantly different.  In brief, we gave it to anyone over 65 as primary prophylaxis or as secondary if the patient had FNE during a previous cycle. The more fulsome answer is we don’t give G-CSF as primary prophylaxis to patients with DLBCL who are under 65 and have no comorbidities that would put them at higher risk for febrile neutropenia, but an episode of FNE as such, or an episode of severe or profound neutropenia sufficiently prolonged in duration to warrant a chemo dose delay or reduction will  trigger the use of G-CSF for subsequent cycles; and comorbidities (such as uncontrolled diabetes, ischemic heart disease or COPD) will prompt the prescribing haematologist to use G-CSF as primary prophylaxis. I assume, but don’t know for certain, that Lamar’s group follows the same guidelines.  In our study 8.5% of 188 received G-CSF. The median age of this group was 70, with 25% of this group under the age of 62, and 25% of this group older than 74.

I have not yet modified the text in the article on this point but will instead await your response to what I have stated above.

-The amount of diabetic patients should be clarified in the whole text. What were the percentages of these patients in the study groups?

I agree with your concern.

In our study 9% of the cohort reviewed did not have blood glucose values and were removed prior to finalizing the cohort to be analyzed (N = 188). For analyses with N = 188, 14.9% had a pre-existing history of diabetes; for analyses with N = 164, 13.4% had diabetes; for analyses with N = 143, 15.4% had diabetes. It's a diagnosis of diabetes prior to the start of the study.  The challenge in putting the figures which speak to the number of study subjects in the hyperglycemic group into the context of the numbers reflecting a pre-existing history of diabetes is that the location, so to speak, of a study subject in the “hyperglycemic” group is based on the trajectory groups, which don't rely on clinical diagnoses. It's whatever the clustering procedure figures out. The two are not comparable.  So, for example, if we had a group of people who were placed in a “diabetic” group based solely on their past medical history, but some of those diabetic patients had very good diabetic control (prior to and  maintained during the study), their glucose values might not land them in the “hyperglycemic”  group as determined by latent group analysis.  I can include explicit comment on this matter in the text if you deem it appropriate. Do you think I have worded things (vide supra) in a way that won’t confuse the reader of the article?

Round 2

Reviewer 1 Report

Comments and Suggestions for Authors

The Authors have addressed all of my concerns with the original manuscript. I have no further comments.